# Identification of a Novel de Novo Variant in the CASZ1 Causing a Rare Type of Dilated Cardiomyopathy

**DOI:** 10.3390/ijms232012506

**Published:** 2022-10-20

**Authors:** Anna Orlova, Daria Guseva, Oxana Ryzhkova

**Affiliations:** 1SRC «Genome», Federal State Budgetary Scientific Institution Research Centre of Medical Genetics, 115522 Moscow, Russia; 2Counselling Unit, Federal State Budgetary Scientific Institution Research Centre of Medical Genetics, 115522 Moscow, Russia; 3Laboratory of Molecular Genetic Diagnosis 3, Federal State Budgetary Scientific Institution Research Centre of Medical Genetics, 115522 Moscow, Russia

**Keywords:** dilated cardiomyopathy, CASZ1 mutation, first de novo mutation

## Abstract

A new de novo frameshift variant has been identified in the CASZ1 gene leading to severe dilated cardiomyopathy. Methods: The proband was analyzed with WES NGS, post-mortem, using dried blood spots on filters. The variant was verified with Sanger sequencing for the proband and her parents. Results: We reported a proband with a new de novo frameshift mutation, c.3781del (p.(Trp1261GlyfsTer29)), in the CASZ1 gene. The clinical presentation was similar to the severe phenotype described in previous studies. Conclusions: In this study, we described a new case with a frameshift mutation in CASZ1 causing a severe phenotype of dilated cardiomyopathy.

## 1. Introduction

Dilated cardiomyopathy (DCM) is the most common hereditary type of cardiomyopathy [1]. DCM is associated with mutations in more than 100 genes encoding elements of the sarcomere and cytoskeleton, including those responsible for nuclear organization, cell energy metabolism, electrolyte homeostasis, gene transcription, and mitochondrial function [1,2].

DCM is one of the leading causes of infant mortality today. About 24% of children who die due to congenital malformations have anomalies in heart development [3]. CHD (congenital heart disease), which includes both the abnormal development of heart structures and large cardiothoracic vessels, is divided into 25 different clinical types. Twenty-one of these denote specific anatomical or hemodynamic lesions, including ventricular septal defect (VSD), atrial septal defect, tetralogy of Fallot, right ventricular double outlet, truncus arteriosus, patent ductus arteriosus, endocardial cushion defect, transposition of the great arteries, aortic stenosis, aortic coarctation, pulmonary artery stenosis, pulmonary atresia, and abnormal connection of the pulmonary veins.

Some of the genes in which mutations lead to DCM are cardiac transcription factors, including homeodomain proteins NKX2-5, GATA family zinc finger proteins GATA4, GATA5, and GATA6, and T-box transcription factors TBX1, TBX5, and TBX20 [4,5].

Previous studies have shown the critical role of transcription factors in fetal heart formation and the development of CHD, including the widely studied zinc finger transcription factor GATA4 and homeodomain transcription factor NKX2-5 [6]. Another transcription factor with zinc finger domains that is highly expressed in cardiac cells during embryogenesis is *CASZ1* [7,8]. *CASZ1* plays an important role in the regulation of the mammalian cardiomyocyte cell cycle and plays an important role in the proper formation of the heart. Genetic fate mapping with an inducible *CASZ1* allele has demonstrated that CASZ1-expressing cells give rise to cardiomyocytes in the first and second heart fields. The generation of a cardiac conditional null mutation shows that CASZ1 is essential for the proliferation of cardiomyocytes in both heart fields and that loss of CASZ1 leads to a decrease in cardiomyocyte cell number. In addition, the loss of *CASZ1* leads to a marked decrease in DNA synthesis, an increase in phospho-RB, and a decrease in the cardiac mitotic index due to the prolongation or arrest of the S phase [7].

In 2006, Lui et al. [9] isolated the expression product of the human *CASZ1* gene, which has two isoforms, CASZ1a and CASZ1b, encoding a protein with 5 and 11 C2H2-type zinc finger motifs, respectively. CASZ5 consists of 1166 amino acids and is a shortened version of CASZ11, consisting of 1759 amino acids. Northern blotting showed that the longer transcript is major. High expression was seen in the human heart, lung, skeletal muscle, pancreas, testis, small intestine, and stomach, but no expression was detected in the brain [9]. The association of loss-of-function mutations in the *CASZ1* gene with increased susceptibility to CHD may be partly ascribed to aberrant heart development. In Xenopus and mammals, *CASZ1* mRNA expression has been found throughout the developing heart, which is critical for proper cardiovascular development [10].

## 2. Case Presentation

### 2.1. Clinical Evaluation

At the age of 3 months of life, dilated cardiomyopathy was diagnosed for the first time. According to ECHO-CG, there was a persistent decrease in ejection fraction (EF) (up to 35–51%), expansion of the cavities of the left and right ventricles, and regurgitation on the tricuspid and mitral valves, and pulmonary artery valve. According to ECHO-CG, at the age of 1 year 10 months, dilated cardiomyopathy with a sharp decrease in the ejection fraction of the left ventricle was revealed. Expansion of the cavity of the left ventricle was noted: KDR—45 mm, KSR—42 mm, EF—12%; left atrium (KDR 25 × 32 mm with spontaneous contrasting symptoms), dilatation of the right atrium, mitral insufficiency of 3 degrees, and aortic insufficiency of 1 degree in the valves.

The girl died at the age of 1 year 10 months with a diagnosis of dilated cardiomyopathy and decompensation, systemic multiple organ failure, post-anoxic brain damage and gastrointestinal and vaginal bleeding.

### 2.2. Genetic Analysis

WES carried out for the proband revealed a novel heterozygous c.3781del (p.Trp1261GlyfsTer29) NM_001079843.3 variant in exon 18 of the CASZ1 gene. Protein synthesis ends at amino acid 1280, which means that part of the functional domains of the long CASZ1a transcript are absent (Figure 1). This variant was not found in the Genome Aggregation Database (gnomAD v.2.1.1 and v.3.1.2) or among samples of 2090 Russian patients’ exomes.

As a result of Sanger sequencing, it was shown that this variant was absent in the proband’s parents. Consequently, we could conclude the de novo origin of the variant in this family. However, we tested the father’s sperm to be sure it was not germinal mosaicism and did not detect the variant, but we have not examined the germ cells of the proband’s mother and, therefore, could not rule out female germinal mosaicism.

The variant was classified as likely pathogenic according to the guidelines for massive parallel sequencing (MPS) data interpretation (criteria PVS1, PM2). The probability of loss-of-function intolerance (pLI; [11]) for CASZ1 in the GnomAD Browser was 1.0 (with 4 observed pLoF SNs vs 60.3 expected SNVs; all observed SNVs are evenly distributed over the gene, falling both in the long- and short-form sequences, and only in the long form have a frequency in the population of no more than 0.00004286 among variants detected both on the exome and on the genome), strongly supporting the fact that this variant is pathogenic. Afterwards, the proband’s parents’ testing variant was classified as pathogenic (criteria PVS1, PM2, PS2). Both paternity and maternity were also confirmed.

## 3. Discussion

Most of the mutations previously described as pathogenic or probably pathogenic in the *CASZ1* gene are associated with neurodevelopmental disorders and are missense variants. Most of them were presented by Edwards JJ et al. in 2020; however, they were characterized only bioinformatically, and family and functional analyses were not carried out [12]. All variants for which such analyses could be performed in their study were inherited from one of the parents and classified as probably benign. In two cases, a recessive type of inheritance is described. In the first case, the p.Ser237Cys mutation in the homozygous state was described in a 6.5-year-old girl from a closely related marriage with DCM and LVNC (left ventricular noncompaction cardiomyopathy), detected at the age of 5 months. The second describes p.Ala1152Thr and p.Thr618Ile variants in a patient with multiple congenital anomalies without intellectual disability, including hyperextensible skin, thoracolumbar scoliosis, and myopathy. Both studies did not prove the causality and pathogenicity of the identified variants; the mutations and the gene were classified as candidates [13,14].

Only three variants with proven pathogenicity (family or functional analysis) have been described (Leu38Pro, Lys351Term, Val815Profs*15) [14,15,16]. All of these are LOF variants located in the coding region of both the short and long forms of the protein and are associated with hereditary heart diseases, such as dilated cardiomyopathy. We have described, for the first time, a variant of the CASZ1 gene in the Russian population with a severe course of the disease, affecting only the long isoform of the protein.

Previous descriptions of the clinical symptoms of patients with pathogenic mutations in the *CASZ1* gene show heterogeneity in the manifestation of the phenotype. Two sources discuss familial cases [14,16]. Xing-Biao Qiu [14] describes a familial case with a proband aged 51 and affected family members aged 25 to 48. The youngest member of the family (the daughter of the proband) was clinically diagnosed in parallel with the discovery of the mutation. However, in addition to DCM, she also had a VSD (ventricular septal defect). The father of the proband was diagnosed with DCM at the age of 43, and died 10 years later from heart failure without a molecular genetic diagnosis. The cause of the disease in this family was the c.1051A>T (p.K351X) mutation (Table 1).

Ri-Tai Huang [16] described another case of a 2-year-old girl with VSD and a family history of the disease. The variant c.113T>C, (p.L38P) was identified according to the results of functional analysis, which significantly reduced the transcriptional activity.

Jun Guo [15] described the frame-shift mutation c.2443_2459delGTGGGCACCCCCAGCCT (p.Val815Profs*14) in an 11-month-old boy with severe heart failure. He died on the second day of hospitalization due to ventricular fibrillation. The researchers ran a familial analysis and showed that it was a de novo mutation and no previous DCM had been observed in the family.

The variety of manifestations of mutations in the *CASZ1* gene is determined by a number of factors, among which both the type of mutation and their localization can occupy an important place.

In 2006, Lui et al. [9] isolated the expression product of the human *CASZ1* gene, which has two isoforms, CASZ1a and CASZ1b, encoding a protein with 5 and 11 C2H2-type zinc finger motifs, respectively. CASZ5 consists of 1166 amino acids and is a shortened version of CASZ11, consisting of 1759 amino acids. Northern blotting showed that the longer transcript was predominant. High expression was seen in the human heart, lung, skeletal muscle, pancreas, testis, small intestine, and stomach, but no expression was detected in the brain [9]. CASZ1a and CASZ1b are equally involved in the transcriptional activation of several target genes expressed during cardiac morphogenesis. At the same time, some researchers noted [14] that CASZ1b is a more evolutionarily conserved isoform and suggested that all further functional analyzes can be performed on it.

As mentioned earlier, the p.(Trp1261GlyfsTer29) variant of the CASZ1 gene does not affect the short isoform of the protein but does affect the long isoform. All previously described pathogenic variants affecting both the long and short forms of CASZ1, according to the Decipher and NMD prediction databases, located in the region predicted by NMD (Figure 2).

The p.Trp1261GlyfsTer29 mutation also falls into the region predicted by NMD, but only in the long transcript (CASZ1b), and it is the only one of those described that falls into the “zinc fingers” domain.

Studies have shown that haploinsufficiency resulting from CASZ1 mutations, regardless of the affected isoform, is potentially a pathological mechanism of DCM [14,17,18].

Nonsense-mediated decay (NMD) is a molecular mechanism whereby potentially defective messenger RNAs (mRNAs) are degraded [19]. NMD has been conventionally regarded as an important mechanism for mRNA quality control. NMD-targeted transcripts (NMDTs) could result from point mutations, insertions/deletions, or alternative splicing events that give rise to a PTC (translation termination codon). NMD is mainly a translation-dependent mechanism. When the translation machinery halts at the first stop codon, and the stop codon is located more than 50–55 nucleotides (NTs) upstream of the last exon–exon junction, the NMD machinery will initiate degradation of the mRNA [20,21]. This stop codon is defined as a PTC. Notably, however, exceptions to this rule have been reported. A transcript may be degraded even when the PTC is located within 50 NTs from the last exon junction, or be resistant to degradation when the PTC is far upstream [19,20,21,22,23,24].

In this study, we present variant c.3781del (p(Trp1261GlyfsTer29)) in the CASZ1 gene as a mutation related to the DCM phenotype. However, our patient had other noncardiac clinical characteristics. As a result of whole-exome sequencing, we did not find variants that are the cause of diseases such as systemic multiple organ failure, post-anoxic brain damage or bleeding. We hypothesize that such a severe clinical picture may be related to the fact that the mutation in the CASZ1 gene that we identified affects only the long transcript of the gene, which is no less conserved than the short one, but is less studied. It is possible to speculate that for the p.Val815Profs*15 and p.Trp1261Term variants located closer to the 3′ end of the gene, early and severe manifestation may be associated not with haploinsufficiency, but with a dominant negative effect of the resulting transcription and translation product. However, we cannot functionally test this assumption, and more research is needed to confirm or refute it.

## 4. Materials and Methods

Clinical data: The patient’s parents were examined at the Research and Counseling Department of the Research Centre for Medical Genetics (RCMG). The patient’s parents provided oral and written consent for this study, and the study was approved by the Ethics Committee of RCMG (approval number 4/1 from 19 April 2021). This initial assessment included a full history of disease and epicrisis post-mortem.

Genetic testing: Whole-exome sequencing was performed for the proband. Blood samples from the unaffected parents were collected, and genomic DNA was extracted using standard methods with the QIAamp DNA Blood Mini Kit (Qiagen, Hilden, Germany). Blood samples from the proband were dried blood spots on the filter that had been stored since 2017; there were no other biomaterials left after the death of the proband. The isolation was carried out according to the manufacturer’s standard protocol, but the first step was the lysis of the filter pieces with blood for 2 hours against the standard 10 minutes at 56 degrees. Then, the standard purification on the column was carried out, and the elution was carried out in a reduced volume of buffer. After DNA fragmentation with the Covaris ME220 Focused-ultrasonicator, DNA quality and concentration were assessed using the 4200 TapeStation. As a result, the patient’s DNA was isolated in a volume of 50 μL at a concentration of 0.794 ng/μL against the standard, which used 50 μL of DNA at a concentration of 5 ng/μL. However, since the protocol allows changing the number of PCR cycles depending on the quality of the DNA, the proband sample was taken in full volume for the reaction. Library amplification was carried out using the KAPA Hyper Prep Kit (Roche), and as a result of 10 amplification cycles instead of 8, a library with a concentration of 35.2 picog/µL was obtained, which was found suitable for further sequencing. The proband’s DNA was analyzed using paired-end reading (2 × 75 bp) on an IlluminaNextSeq 500 sequencer (Illumina, San Diego, CA, USA). Target enrichment was performed with an IDT xGen^®^ Exome Research Panel v.1 solution capture array, including the coding regions of 19,396 known genes. Sequencing data were processed using a standard computer-based algorithm from Illumina software, presented on the https://basespace.illumina.com website (Enrichment, version 3.1.0, Illumina, San Diego, CA, USA). Average coverage for this sample was 57.4×, coverage with (10×)–99.5108%, and coverage with (20×)–97.4484%. The sequenced fragments were visualized using Integrative Genomics Viewer (IGV) software (©2013–2018 Broad Institute, Cambridge, MA, USA and the Regents of the University of California, Oakland, CA, USA).

A total of 180,627 variants were identified, with 41,127 of these passing the pass-filter. Filtering of the variants was based on their frequency—less than 1% in gnomAD—and coding region sequence effect, such as missense, nonsense, coding indels, and splice sites. A total of 82 variants were found that had been previously associated with disease (OMIM, ClinVar, and HGMD) and were truncating variants. Of these, only one variant was in a gene whose mutations have been described in similar phenotypes (CASZ1). The remaining variants were in recessive genes or in genes that did not explain the clinical diagnosis of the patients. The remaining 379 variants were also analyzed to rule out another possible cause of the disease including noncardiac characteristics. Of these, two variants in the same gene and variants in the dominant genes that would lead to abnormalities were not found. Bioinformatics tools BayesDel_addAF, DANN, DEOGEN2, EIGEN, FATHMM-MKL, LIST-S2, M-CAP, MutationTaster, SIFT, NetGene2 and Splice_ai were used to predict the potential impact of the identified mutations on protein function. The variants’ clinical significance was evaluated according to the guidelines for massive parallel sequencing (MPS) data interpretation [25].

Amplification and Sanger sequencing were performed to validate the exome variant in the *CASZ1* gene in the proband and its absence in the parents. Amplifications were carried out by PCR with Taq polymerase in a Veriti Dx Thermal Cycler (Thermo Fisher Scientific, Waltham, MA, USA). The protocol used for amplification included the following steps: 95 °C for 5 min; 32 cycles of 94 °C for 45 s; 62 °C for 45 s; 72 °C for 45 s; 72 °C for 5 min; 4 °C holds. Automatic Sanger sequencing was carried out using an ABIPrism 3100xl Genetic Analyzer (Applied Biosystems, Foster City, CA, USA) according to the manufacturer’s protocol. Sequencing results were analyzed using Chromas (Technelysium Pty Ltd., South Brisbane, Australia). To amplify the fragment encompassing the candidate variants, custom primers were used (according to the NM_001079843.3 reference sequence): CASZ1_18F: CAGGCAGGAGCTCTTGAAAGTC, CASZ1_18R: GGCCATGGTCTTGATGCCTGC.

## 5. Conclusions

When analyzing the causality and pathogenicity of mutations in the *CASZ1* gene, as well as in its further studies, it is necessary to focus not only on the short but also on the long form of the protein. The wide clinical variety of manifestations and severity of the course of disease, as well as the possible presence of a recessive type of inheritance, should be taken into account.

## Figures and Tables

**Figure 1 ijms-23-12506-f001:**
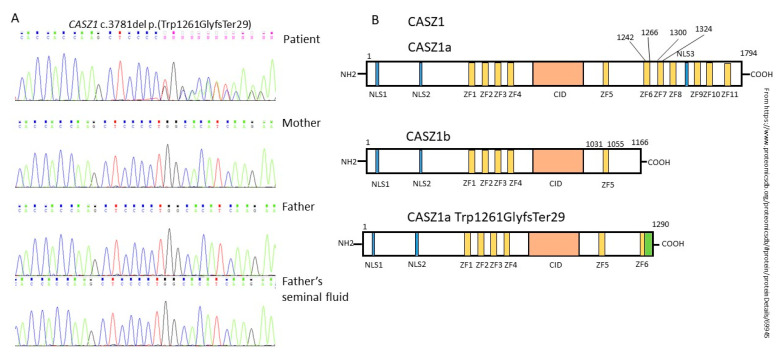
(**A**) Sanger sequence validation of proband and parents; (**B**) structure of the CASZ1 protein (isoforms CASZ1a, CASZ1b, and mutant protein).

**Figure 2 ijms-23-12506-f002:**
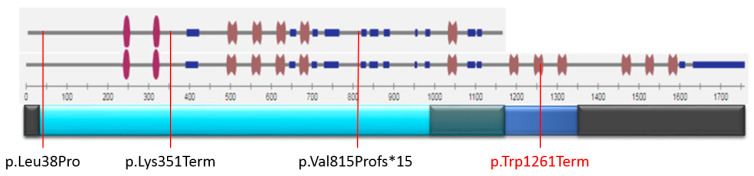
Domains of both CASZ1 isoforms and the NMD area with an indication of mutations on it. Numbers—base sequence; 
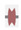
—zinc finger domain; 
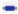
—low-complexity regions; 
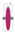
—internal repeats; 
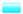
—both CASZ1a and CASZ1b sequences (NMD region); 
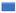
—CASZ1b sequence (NMD region); 
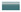
—CASZ1a non-NMD region; 
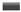
—CASZ1b non-NMD region; 

mutation location.

**Table 1 ijms-23-12506-t001:** Summary table with parameters of patients and their affected relatives.

Family	Affected Member of Family (Male)	Age (Enrollment/Initial Diagnosis of DCM)	Diagnosis ^1^	LVEF %	LVFS %	LVESD mm	LVEDD mm	Another Symptoms ^2^	Mutation	Ref.
1	Proband (m)	51 / 41	DCM	30	17	63	76		c.1051A > T, p.K351X	Xing-Biao Qiu [14]
	Brother	48 / 45	DCM	32	20	43	54			
	Sister	39 / 33	DCM	30	14	49	55			
	Daughter	25 / 25	DCM, VSD	39	23	44	58			
2	Proband (f)	2	VSD, DORV	-	-	-	-		c.113T > C, L38P	Ri-Tai Huang [16]
	Grandfather	59	VSD	-	-	-	-			
	Father	31	VSD	-	-	-	-			
	Aunt	26	VSD							
3	Patient	11 mo/11 mo	DCM, LVNC	25	-	-	51	Mild mitral regurgitation, pneumonia, died due to VF	c.2443_2459del17, p.Val815Profs*14	Jun Guo [15]
4	Patient *	1y10 mo/3 mo	DCM	12	-	42	45	Mild mitral and aortal regurgitation, died due to MODS		*

^1^ DCM—dilated cardiomyopathy; VSD—ventricular septal defect; DORV—double outlet right ventricle; LVNC—left ventricular noncompaction cardiomyopathy. ^2^ VF—ventricular fibrillation; MODS—multiple organ dysfunction syndrome. * Current study.

## Data Availability

The data presented in this study are available in the article.

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
