# Peer review of "Identification of a Novel de Novo Variant in the CASZ1 Causing a Rare Type of Dilated Cardiomyopathy"

_ijms, 2022, doi:10.3390/ijms232012506_

Round 1
Reviewer 1 Report
The study of Orlova et al. is interesting because it presents a novel mutation in
CASZ1 which causes dilated cardiomyopathy (DCM). This is the 4th variation in the gene to be associated with DCM, the novelty is in the mutation being present in the splice variant of the gene not yet reported to associate with disease.
There are several points that need to be addressed:
1. The introduction has to be improved.
A.The 3rd paragraph stating:”Among the genes in which mutations lead to DCM, most are cardiac transcription factors …”, is not correct. Many other genes, mainly contributing to force generation were found to contribute to more to DCM.
B. The information on CHD (congenital heart disease), is missing a reference.
C. The information on the function and the splicing forms of CASZ1 should be presented in the introduction.
2. The discussion: A. the paragraph starting with “Only three variants with proven pathogenicity”, line 107 is missing a reference.
B. The numbering of the references in Table 1 is not correct.
C. The paragraph starting: “All previously described pathogenic variants affecting both the long and short forms of CASZ1 …” line 157, is unclear, and where does this information come from?
D. The same also refers to the following paragraph:” However, it is also known that the NMD effect is not characteristic of all variants that lead to a shortening of the open reading frame (ORF),…”, line 168.
3. The legends for Figures 2 and 3 are not detailed enough to be able to understand what they represent.
4. The method section should include the bioinformatics analysis and how other variants were excluded.
Reviewer 2 Report
Reviewer report manuscript Orlova et al, “Identification of a novel de novo variant in the CASZ1 causing a rare type of cardiomyopathy”
In this manuscript the authors describe the discovery of a de novo loss-of-function mutation in the CASZ1 gene, identified performing WES analyses, in a young child that suffered and died (age 1 year, 10 months) from dilated cardiomyopathy and multiple cardiac valve problems. Clinical presentation in the child, as well as an overview of that in other patients with CASZ1 variants is presented.
Main comment:
1. The authors basically only present the identification of a de novo mutation possibly involved in pediatric DCM: more information should be included (see also below).
2. the de novo character of this variant (and it’s absence from the general population) underscores it’s involvement in the cardiac problems in this child. However, as the clinical presentation is complex, in order to judge whether this is the (only) causal variant, presenting the filtering strategy, including how many variants were discarded with each step, as well as which other putative candidates remained at the end of such filtering strategy, is warranted. May there have been more interesting candidates, putatively also explaining clinical characteristics other than cardiomyopathy and what about their putative de novo status?
3. The authors present this as a mutation related to a/the DCM phenotype, however other clinical characteristics (and even non-cardiac characteristics in the end) are also presented. Nevertheless, little effort is taken to explain why the mutation is particularly associated to DCM and it’s putative role in the other manifestations is underrepresented. Elaborating on that would be recommended.
4. Related to the above: please explain more clearly that the focus is on childhood DCM and in which genes mutations may be expected. Now, the focus is mainly on transcription factor genes and their role in pediatric cardiomyopathy, in particular in the introduction, but also many other genes as well as “adult” (D)CM genes have shown to be involved. This should be reflected upon in the manuscript.
5. Although providing information on other CASZ1 mutation helps in supporting the putative role of the one identified in this study, it does not seem necessary to provide the details in the form of Table 1, presenting this shortly in the text of the discussion would suffice.
Other/minor comments:
Abstract:
-What do the authors refer to with “hard” in the “hard phenotype” (last 2 sentences of the abstract? Please clarify.
Introduction:
-I would recommend to save most of the information on CASZ1 given at the end of the introduction (last 2 paragraphs of the introduction) for the discussion.
Results:
-The authors state/suggest that due to the identified mutation protein synthesis ends at amino acid 1280 (line 71-72, page 4), however in most cases, unless the mutation is in the last exon or the last 50 nt’s of the forelast exon, the general concept is that this leads to the protein not being produced at all due to NMD. Please indicate this (and as shown by mentioning the pLI value in gnomAD, the authors are aware of this). In addition, unless proof of its existence would have been presented, remove the “mutant protein” from Fig 1B, as that most likely is not produced at all. Instead indicate the position of the mutation at the representation of the protein product of isoform CASZ1a.
-The authors apply rule PS2 of the ACMG criteria, however do not show/indicate that both paternity and maternity was confirmed, so rule PM6 should be applied.
Discussion:
-Please reflect on what would be the reasons for the L38P leading to only VSD and why the p.351* does not result in severe disease at young age. This may indeed relate to a putative dominant-negative effect, however please present this with more care, unless data exist to support this. Related to that it is recommended to not include Figure 2.
-Related to the above: the discussion on which isoform is the most predominant one, more conserved, involved in transcription is also confusing: what would be most relevant from the referred studies for the finding now presented?
-It is unclear how results in ref’s 18 and 19 would show that LoF is the pathological mechanism in DCM. Instead, I would recommend to here present the data of the role of CASZ1 in cardiac development that is now already provided in the introduction (see also above).
Conclusions:
-Maybe this is a general part of manuscripts in this journal, however I would recommend to omit this as a separate thing and just add this to the discussion, including a sente4nce summarizing the main result of this manuscript.
Round 2
Reviewer 1 Report
The issues were adequately addressed.
Author Response
Thank you very much for your time and your corrections!
Reviewer 2 Report
The authors significantly improved their manuscript, which would now be ready for publication, exceptfor one comment not being satisfactorily adressed:
Point: I would recommend to save most of the information on CASZ1 given at the end of the introduction (last 2 paragraphs of the introduction) for the discussion.
Response: We think it is right to give brief information about the gene in the introduction, so that later on when reading the article the reader will have an idea of what the article is about. We tried to present the information as briefly as possible. However, we have added more information about the function of the gene and transcripts. The added text is highlighted in green.
I would still recommend to only briefly touch upon this in the introduction and elaborate on this in the discussion to take care the reader is also interested in reading the full manuscript.
Author Response
Thank you very much for your time and your corrections!
Point: I would still recommend to only briefly touch upon this in the introduction and elaborate on this in the discussion to take care the reader is also interested in reading the full manuscript.
Response: Thank you very much for your point! Another reviewer on the contrary asked to increase this part. We don’t think the introduction should contain the detailed description, but we will keep the short review.
Also we had checked and improved some spell mistakes. Thank you for your attention!